# Characteristics and clinical outcome in 312 patients with moderate to severe pneumonia due to SARS-COV-2 and hyperinflammation treated with anakinra and corticosteroids: A retrospective cohort study

Ismael Francisco Aomar-Millán [1,2]☯ *, Javier Martínez de Victoria-Carazo[1], Daniel Fernández Reyes[1], Úrsula Torres-Parejo[3], Laura Pérez Fernández[4], Silvia Martínez-Diz[5], Angel Ceballos Torres[1], Jairo López Gómez[1], Francesco Bizzarri[6], Enrique Raya Álvarez[2,6], Juan Salvatierra[2,6]☯

**1** Department of Internal Medicine, Hospital Universitario Clínico San Cecilio, Granada, Spain, **2** Biosanitary Research Institute of Granada, ibsGRANADA, Granada, Spain, **3** Department of Statistics, Faculty of Health Sciences, University of Granada, Granada, Spain, **4** Zubia Healthcare Center, Granada, Spain, **5** Department of Preventive Medicine, Hospital Universitario Clínico San Cecilio, Granada, Spain, **6** Department of Rheumatology, Hospital Universitario Clínico San Cecilio, Granada, Spain

☯ These authors contributed equally to this work.

* iaomarmillan@hotmail.com

## Abstract

### Objective

To assess the clinical outcome (death and/or Intensive Care Unit (ICU) admission) based on the time from hospital admission to the administration of anakinra and the possible usefulness of a "simplified" SCOPE score to stratify the risk of worse prognosis in our cohort of patients with moderate/severe SARS-CoV-2 pneumonia, both vaccinated and unvaccinated, that received anakinra and corticosteroids. In addition, the clinical, analytical, and imaging characteristics of patients at admission are described.

### Methods

Retrospective cohort study of 312 patients admitted to Hospital Clínico San Cecilio in Granada for moderate/severe pneumonia caused by SARS-CoV-2 that received anakinra and corticosteroids between March 2020 and January 2022. Clinical and analytical data were collected as well as the patient outcome at 30 and 60 days after admission. Three treatment groups were established according to the time from hospital admission to administration of anakinra: early (1st–2nd day), intermediate (3rd–5th day), and late (after the 5th day).

### Results

The median age was 67.4 years (IQR 22–97 years) and 204 (65.4%) were male. The most common comorbidity was hypertension (58%). The median time from the start of symptoms to anakinra administration was 6 days (IQR 5–10) and the SaFi ($SaO_2$/$FiO_2$) was 228 (IQR

**Data Availability Statement:** All relevant data are within the paper and its Supporting Information files.

**Funding:** The authors have received funding from SOBI for the translation into English of the present manuscript. The funders had no role in study design, data collection and analysis, decision to publish, or preparation of the manuscript.

**Competing interests:** Ismael F. Aomar Millán has participated on Advisory boards of Ferrer Internacional, Sanofi, SOBI, Astra Zeneca, and GSK. Juan Salvatierra has participated on Advisory boards and Presentations of Roche, SOBI, Novartis, Lilly, GSK, and Pfizer. Enrique Raya has participated on Advisory boards and received study funds from Novartis, Roche, Lilly, GSK, and Pfizer. The remaining authors declare no conflict of interests. This does not alter our adherence to PLOS ONE policies on sharing data and materials.

71–471). The cure rate was higher in the early-onset anakinra group versus the late-onset group (73% vs 56.6%). The latter had a higher percentage of deaths (27.4%) and a greater number of patients remained hospitalized for a month (16%). On admission, the patients had elevated C-reactive protein (CRP), ferritin, and D-dimer values and decreased total lymphocytes. Analytical improvement was observed at both 72 hours and one month after treatment. 42 (13.5%) required ICU admission, and 23 (7.3%) orotracheal intubation. At 60 days, 221 (70.8%) were discharged, 87 (27.8%) had died and 4 (1.4%) remained hospitalized. The mean dose of anakinra was 1000 mg (100–2600 mg) with differences found between the dose administered and the clinical outcome. There were no differences in the primary outcome based on vaccination.

A simplified SCOPE score at the start of anakinra administration was lower in patients with better clinical evolution.

## Conclusions

Early treatment with anakinra and corticosteroids was associated with a better outcome regardless of vaccination status. A simplified SCOPE was found to be a good prognostic tool.

## Introduction

In March 2020, the World Health Organization (WHO) declared a pandemic caused by SARS-CoV-2 [1, 2], and by November 2022 there were more than 600 million confirmed cases worldwide and more than 6.5 million deaths [3].

The syndrome associated with this infection presents several clinical manifestations with a direct correlation between the severity of pneumonia, systemic inflammation, progression to respiratory failure and death [4].

Some patients with COVID-19 develop a hyperinflammatory phase with clinical deterioration that may lead to acute respiratory distress syndrome (ARDS), the leading cause of death [5]. The pathogenesis of lung injury by SARS-CoV-2 involves direct viral damage and dysregulation of the immune response involving a multi-system inflammatory reaction and significant release of cytokines [6, 7].

Regarding laboratory tests, it is characterized by a marked increase in acute phase reactants such as C-reactive protein (CRP), ferritin, and D-dimers (DD), as well as a significant decrease in T lymphocytes [8].

Once the hyperinflammatory phase is triggered, prompt and individualized treatment is essential to control it and prevent ARDS and death. In severe SARS-CoV-2 infection, an increase in the expression of IL-1α and IL-1β has been shown to occur before the deterioration of respiratory function, supporting the involvement of IL-1 in the pathophysiology of ARDS in these patients [9]. Anakinra is a recombinant IL-1 receptor antagonist that is commonly used to treat hyperinflammatory conditions and, therefore its use in the hyperinflammatory phase of moderate/severe SARS-CoV2 infection improves clinical symptoms and survival by blocking IL-1α and IL-1β controlling the altered regulation of cytokine responses [10].

During the pandemic, corticosteroids [11–13] and various immunomodulators, such as tocilizumab or baricitinib [14, 15], have proven useful for controlling the inflammatory phase of SARS-CoV2 infection. In December 2021, following the SAVE-MORE clinical trial [16], the

European Medicines Agency approved anakinra for the treatment of patients with moderate/ severe SARS-CoV-2 pneumonia at high risk of progression to severe respiratory failure based on plasma levels greater than 6 ng/mL of soluble urokinase plasminogen activator receptor (suPAR). This trial demonstrated a 64% reduction in clinical worsening and a 55% reduction in death on day 28, as well as an 80% reduction in patients with laboratory criteria for cytokine storm at the time of treatment initiation. Recently, the Severe Covid Prediction Estimate (SCOPE) calculated from circulating concentrations of CRP, DD, ferritin and IL-6 has been validated to stratify the risk of progression to severe respiratory failure or death with a similar predictive accuracy to suPAR [17]. The SCOPE establishes a cut-off point of 7 points to stratify patients at risk of progressing to severe respiratory failure or death.

Before the SAVE-MORE trial, several studies had suggested the benefits of anakinra as a treatment for patients with hyperinflammation associated with severe SARS-CoV-2 infection, both in its initial phases [18–21] and as rescue treatment after failure of other immunomodulators [10].

In the present study, our objective was to assess the clinical outcome (death and/or ICU admission) based on the time from hospital admission to the administration of anakinra (early: starting on days 1–2; intermediate: between days 3–5; late: after the 5th day of admission) in our cohort of patients with moderate/severe SARS-CoV-2 pneumonia, both vaccinated and unvaccinated, that received anakinra and corticosteroids during the pandemic, as well as the possible usefulness of a "simplified" SCOPE score to stratify the risk of worse prognosis. In addition, the clinical, analytical, and imaging characteristics of patients at admission are described and whether prior vaccination influenced the clinical response to treatment with anakinra.

## Material and methods

### Study design and population

In this retrospective cohort study, we analyzed all patients, 18 years and over, admitted to the Hospital Clínico San Cecilio in Granada (Spain) with moderate/severe pneumonia due to SARS-CoV-2, which was confirmed by PCR via nasopharyngeal exudate, and who received anakinra and corticosteroids as an immunomodulator treatment between 15 March 2020 and 31 January 2022.

### Study procedures and data collection

The clinical and analytical data of the included patients were extracted from the electronic medical records of our center's database (DIRAYA) and fully anonymized before entering into a database in SPSS version 23 so ethics committee waived the requierement for informed consent. Patient data were collected using the Case Report Form (CRF) following the recommendations of the ISARIC-WHO COVID-19 CRF. The following data were collected from the medical records of patients admitted to our center between 15 March 2020 and 31 January 2022: demographic data, clinical characteristics, severity scales at admission, comorbidities, $SaFiO_2$ on admission and at the start of treatment with anakinra, laboratory parameters on admission, before treatment with anakinra, at 72 h and one month after treatment, radiological findings on computed tomography and/or chest X-ray, time from onset of symptoms and admission to the administration of anakinra, the dose of anakinra used, previous treatments administered, previous vaccination and clinical outcome (discharge, admission to the ICU, and/or death). A "simplified" SCOPE score was calculated based on serum CRP, ferritin and DD levels without taken IL-6 into account since it is not a routine determination in most hospitals. The laboratory tests were performed at our hospital. The COVID-19 Data and

Reporting System (CO-RADS) score was used to identify the probability of COVID-19 infection based on a CT scan [22] and extent according to the guidelines of the British Society for Thoracic Imaging (BSTI) classification system [23]. Any missing data are noted in the tables.

All patients received standard treatment according to the protocol of the multidisciplinary committee of our center in each wave of the pandemic.

## Primary outcomes

Characterization of the patients, including demographic and clinical data, comorbidities, time from symptom onset and admission to treatment administration, SaFi on admission and before the administration of anakinra, "simplified" SCOPE score, laboratory parameters before treatment, at 72 hours and at one month, radiological data of COVID-19, severity of the disease and clinical outcome (cure, ICU admission, and/or death).

## Definitions

Patients with moderate and severe disease were defined according to the WHO definition of disease severity for COVID-19. Severe pneumonia was determined as follows: baseline oxygen saturation <93% or O2 partial pressure <65 mmHg with radiological evidence (chest X-ray or computed tomography) of unilobar or multilobar involvement consistent with COVID-19. Hyperinflammation was defined as the presence of normal procalcitonin with two or more of the following criteria: CRP > 100 mg/L, ferritin level > 500 μg/L, DD level > 0.5 mg/L.

The severity of unilobar or multilobar lung involvement was assessed by chest X-ray and/or computed tomography. A semi-quantitative scoring system developed by the British Thoracic Imaging Society [23] was used to estimate lung involvement based on the area affected, and was classified as mild (<25%), moderate (25–50%), or severe (> 51%).

Age was stratified into two subgroups: <65 years and >65 years. The cohort was divided into three groups based on the time from admission to the administration of anakinra (on the 1st–2nd day, between the 3rd–5th days, and after the 5th day) and regarding the clinical outcome at 30 days (discharge without ICU admission, discharge after ICU admission, and death). The start of the illness was defined as the first day of symptoms. Obesity was defined as a body mass index (BMI) greater than 30 kg/m$^2$ and renal failure at admission as an eGFR < 60 ml/min present at least 3 months prior to admission.

A "simplified" SCOPE score was calculated based on CRP, ferritin, and DD levels (Fig 1).

The standard treatment protocol of our hospital during the first wave of the pandemic included the use of weight-adjusted methylprednisolone with progressive reduction for 15 days (1–2 mg/kg/day of IV MTPN for 3–5 days; subsequently, 40 mg/day for 3 days, 30 mg/day for 3 days, 20 mg/day for 3 days, and finally 10 mg/day for 3 days) together with hydroxychloroquine (800 mg/day the first day and 400 mg/day for 4 more days), azithromycin (500

| D-dimers (mg/l) | CRP (mg/l) | Ferritin (ng/ml) | Points |
|---|---|---|---|
| 0.1–0.4 | 0.3–25.0 | 10–225.0 | 0 |
| >0.4-0.57 | >25.0-45.0 | >225.0-450.0 | 1 |
| >0.57-0.90 | >45.0-85.0 | >450.0-750.0 | 2 |
| >0.90 | >85.0 | >750.0 | 3 |

**Fig 1. A simplified SCOPE score.** CRP: C-Reactive protein. Each of the three biomarkers is allocated 0 to 3 points according to the concentration. The final score is the addition of the points provided by each biomarker.

mg on day 1 then 250 mg/day for 5 days), lopinavir/ritonavir (800/200 mg daily for 14 days) and ceftriaxone (2 g daily for 7–10 days) along with thromboembolism prophylaxis with bemiparin at a dose adjusted for thrombotic risk (low risk 3,500 IU/day, intermediate risk 5,000 IU/day). Likewise, at the discretion of the responsible physician, the use of tocilizumab and/or anakinra was allowed.

After the publication of the Recovery study, our hospital treatment protocol was changed and weight-adjusted methylprednisolone was replaced with dexamethasone 6 mg/day for 10 days in addition to standard treatment that included remdesivir if criteria were met and bemiparin at therapeutic doses. Treatment with tocilizumab and/or anakinra was also allowed according to the characteristics and course of the patients.

## Ethical considerations

This study was conducted in accordance with the principles of the Helsinki declaration and received a favorable opinion from the Granada Province Research Ethics Committee/Drug Research Ethics Committee. Code number 1329-N-21.

## Statistical analysis

After the descriptive analysis, the differences between the variables studied were assessed. Qualitative variables were assessed using the Chi-square test to compare proportions. For quantitative variables, normality was tested using the Kolmogorov-Smirnov test. The Mann-Whitney U test was used in cases of non-normality and the Analysis of Variance of a Factor (ANOVA) for normally distributed variables. Two Cox regressions were performed to calculate the Hazard Ratio separately for mortality and medical discharge according to the time from hospital admission to the administration of anakinra which was corrected for age, gender and CRP at admission. The Kaplain- Meier curves were added for probability of medical discharge due to recovery. The Log-Rank statistical test was used to establish whether there were significant differences between the curves. The statistical software used was SPSS version 23.

## Results

### Demographic characteristics, comorbidities, severity scales at admission, and treatment before anakinra administered

A final total of 312 patients admitted to the Hospital Clínico San Cecilio in Granada (Spain) with moderate/severe pneumonia due to SARS-CoV-2 confirmed by PCR via nasopharyngeal exudate and who received anakinra and corticosteroids as an immunomodulator treatment between 15 March 2020 and 31 January 2022, were included.

The comorbidities and scores on the severity scales qSOFA and CURB 65 at admission in the different treatment groups are shown in Table 1. The median age was 67.4 years (IQR 22–97 years) and 204 (65.4%) were male. The most frequently observed comorbidities were hypertension (58%), ischemic heart disease (91.7%), heart failure (87.5%), chronic obstructive pulmonary disease (COPD) (78.5%), obesity (60.3%), diabetes mellitus (68.8%) and renal failure (86.2%). However, significant differences were only found in the percentage of heart failure between treatment groups 2 and 3 (p-value 0.04).Regarding the severity scales on admission, the most frequently observed score was 1 on the CURB-65 scale (45.2%) and 0 on the qSOFA scale (168, 59.6%) with no differences between the treatment groups.

Anakinra was administered later in older patients and in those who had previously received tocilizumab.

**Table 1. Comorbidities and severity scales at admission for the different treatment groups.**

| COMORBIDITIES | Group I (n = 52) | Group II (n = 153) | Group III (n = 107) | p-value (groups 1–2) | p-value (Groups 1–3) | p-value (Groups 2–3) |
|---|---|---|---|---|---|---|
| | n (%) | n (%) | n (%) | | | |
| Hypertension | 30 (57.7) | 86 (56.2) | 65 (60.7) | 0.852 | 0.712 | 0.465 |
| COPD/Asthma | 9 (17.3) | 29 (19.0) | 29 (27.1) | 0.792 | 0.174 | 0.120 |
| Obestiy | 24 (46.2) | 59 (38.6) | 41 (38.3) | 0.335 | 0.346 | 0.968 |
| Cardiac ischemia | 4 (7.7) | 14 (9.2) | 8 (7.5) | 0.748 | 0.961 | 0.633 |
| Heat Failure | 6 (11.5) | 14 (9.2) | 19 (17.8) | 0.616 | 0.312 | 0040* CI (-0.17;-0.01) |
| Diabetes | 14 (26.9) | 44 (28.8) | 40 (37.4) | 0.800 | 0.191 | 0.143 |
| Renal insufficiency | 7 (13.5) | 16 (10.5) | 20 (18.7) | 0.553 | 0.410 | 0.059 |
| Pluripathology | 6 (11.5) | 15 (9.8) | 14 (13.1) | 0.722 | 0.783 | 0.408 |
| SEVERITY SCALES | $\bar{x} \pm SD$ | $\bar{x} \pm SD$ | $\bar{x} \pm SD$ | p-value (groups 1–2) | p-value (Groups 1–3) | p-value (Groups 2–3) |
| qSOFA score | 0.44 ± 0.57 | 0.48 ± 0.59 | 0.40 ± 0.56 | 0.710 | 0.057 | 0.302 |
| CURB65 score | 0.79 ± 0.78 | 0.94 ± 0.82 | 1.05 ± 0.81 | 0.241 | 0.674 | 0.305 |

COPD: chronic obstructive pulmonary disease. qSOFA: Quick Sepsis-related Organ Failure Assessment. CURB65: Pneumonia Severity Score.

Treatment prior to administering anakinra consisted of corticosteroids in 277 (88.8%) patients and corticosteroids plus tocilizumab in 35 (11.2%). Another 43 patients (13.8%) received treatment with remdesivir. There were no differences in clinical outcome based on the previous treatment received (p-value 0.075) or with remdesivir treatment (p-value = 0.39).

When analyzing the Kaplan Meier survival curves, we observed that patients older than 65 years as well as the presence of hypertension, heart failure and/or diabetes had a lower accumulated survival probability than those who did not suffer from these comorbidities.

## Symptoms and SaFi

On admission, the duration from the onset of symptoms was between 6 and 11 days (IQR 5–10) for all patients. The most frequently observed symptoms were fever, cough and dyspnea followed by tachycardia, myalgia, asthenia, ageusia, anosmia and diarrhea.

The mean SaFi (SaO2/ FiO2. Relationship between transcutaneous oxygen saturation and fraction of inspired oxygen) on admission was 399 (87–476) and at the time of anakinra administration it was 228 (71–471). There were significant differences between the mean SaFi at admission according to the primary outcome at 60 days, being higher for those patients who were discharged without admission to the ICU compared to those who required admission to the ICU or died (p-value = 0.000) (Table 2).

There were also significant differences between the 60-day outcome and SaFi at the start of anakinra treatment. The administration of anakinra in patients with a higher SaFi was associated with a 60-day higher percentage of medical discharges without ICU admission compared to those who died or were discharged after ICU admission (p-value = 0.008) (Fig 2).

Furthermore, patients that started anakinra treatment during the first two days of admission showed a higher mean SaFi than those that started it later (p-value 0.001).

## Laboratory, vaccine, and radiological data

On admission, CRP, ferritin, and DD were significantly increased and the total lymphocyte count decreased. Both at 72 h and one month after the start of anakinra treatment, a significant decrease in CRP, ferritin and DD was observed, as well as an increase in total lymphocytes (Table 3).

**Table 2. Clinical outcome at 60 days based on SaFi, anakinra and a simplified SCOPE.**

| | Discharge without ICU n = 191 (61.2%) | Discharge with ICU n = 30(9.6%) | Death n = 87 (27.9%) | p-value (Discharge without ICU / Discharge with ICU) | p-value (Discharge with ICU / Death) | p-value (Discharge with ICU / Death) |
|---|---|---|---|---|---|---|
| SaFi on admission (mean; SD) | 415.25; 57.59 | 366.03; 119.36 | 376.84;97.39 | 0.000** | 0.622 | 0.000** |
| SaFi at the start of treatment with anakinra (mean; SD) | 262.40; 100.29 | 212.30;94.12 | 163.09;83.49 | 0011* | 0.008** | 0.000** |
| Total no. days Anakinra treatment (Mean; SD) | 11.92; 8.94 | 20.48;11.26 | 10.75;11.12 | 0.000** | 0.000** | 0.352 |
| Total Anakinra dose (Mean; SD) | 1136.13;481.89 | 516.67;545.25 | 822.99;605.54 | 0.000** | 0016* | 0.000** |
| A simplified SCOPE (mean SD) | 6.32;1.92 | 6.83;2.04 | 6.91;1.71 | 0.179 | 0.846 | 0020* |

ICU: Intensive Care Unit.

All patients had a chest X-ray and multilobar involvement was observed in 97.1%. In addition, 112 had a chest CT with 17 (15.1%) showing moderate lung involvement, 46 (41.1%) moderate-severe and 49 (43.8%) severe. Most patients (94.5%) had a CO-RADS score of 5 at the time of admission.

Because our data collection end date was 31 January 2022, only 35 patients (11.2%) in our cohort had been vaccinated against SARS-CoV-2. The most frequent vaccination regimen was the administration of 2 doses of Pfizer BioNTech in 16 (45.7%) patients. There were no differences between treatment with anakinra, clinical outcome, and previous vaccination (p-value = 0.301) (Table 3).

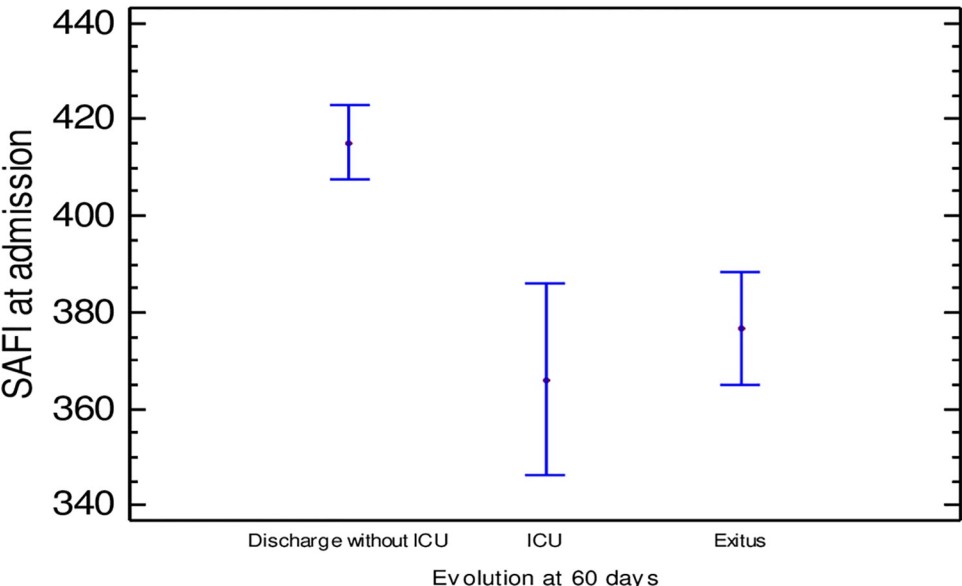

**Fig 2. Relationship between mean SaFi at admission and clinical outcome at 60 days.**

**Table 3. SaFi, laboratory, radiology, vaccine status, and inpatient treatment data.**

| Data on admission | Global | Group 1: Anakinra in the first 2 days | Group 2: Anakinra from the 3rd to 5th day | Group 3: Anakinra after the 5th day | p-value (group 1–2) | p-value (group 2–3) | p-value (group 1–3) | Missing values n (%) |
|---|---|---|---|---|---|---|---|---|
| SaFi on admission (mean) | 399.19 | 377.40 | 400.02 | 408.60 | 0.090 | 0.398 | 0.024 | 0(0) |
| SaFi at the start of treatment with anakinra (mean) | 228.60 | 271.73 | 224.78 | 213.09 | 0006** | 0.357 | 0001** | 0(0) |
| CRP (mg/L), mean (median; SD) | 105.75 (101; 75.66) | 140.83 (131; 82.93) | 109.16 (102; 70.60) | 83.41 (77.5; 71.87) | 0.009** | 0.005** | 0,000** | 10 (3.2) |
| Ferritin (mg/L) mean (median; SD) | 1023.33 (771.5; 837.68) | 1136.12 (776; 1006.46) | 1020 (778; 824.47) | 971.30 (742; 763.52) | 0.413 | 0.634 | 0.256 | 10 (3.2) |
| D-dimer (mg/L) mean (median; SD) | 6.09 (1.0; 52.59) | 1.73 (1.0; 4.76) | 8.56 (0.9; 73.70) | 4.82 (1.37; 20.17) | 0.506 | 0.614 | 0.279 | 10 (3.2) |
| Total lymphocytes (x10^9) mean (median; SD) | 1084.32 (790; 2881.35) | 869.81 (840; 398.19) | 989.93 (795; 99.36) | 1322.27 (740; 4747.90) | 0.401 | 0.415 | 0.494 | 12 (3.8) |
| Simplified SCOPE, mean (SD) | 6.51 (7; 1.90) | 6.9 (7; 1.92) | 6.44 (7; 1.84) | 6.41 (6; 1.96) | 0.123 | 0.918 | 0.141 | 10 (3.2) |
| Degree of multilobar involvement on chest X-ray n (%) | 303 (97.1) | 52(100) | 150 (98.0) | 101 (94.4) | 0.727 | 0.216 | 0.195 | 0 (0) |
| CO-RADS 5 on CT Chest n (%) | 49 (15.7) | 11 (21.2) | 23 (15.0) | 15 (14.0) | 0.305 | 0.820 | 0.254 | 3 (1.0) |
| **Laboratory data at 72 hours** | | | | | | | | |
| CRP (mg/L) mean (median; SD) | 43.49 (20; 56.16) | 47.39 (24.4; 50.65) | 43.22 (22; 55.69) | 41.84 (16.05; 59.89) | 0.637 | 0.854 | 0.569 | 17 (5.4) |
| Ferritin (mg/L) mean (median; SD) | 1005.18 (752.5; 841.78) | 1036.71 (818.5; 903.64) | 992.61 (710.5; 843.07) | 1006.63 (783; 814.53) | 0.752 | 0.897 | 0.836 | 18 (5.8) |
| D-dimer (mg/L) mean (median; SD) | 6.30 (1.2; 48.31) | 3.13 (1.14; 7.02) | 9.09 (1.12; 68.63) | 3.98 (1.43; 7.61) | 0.533 | 0.458 | 0.551 | 20 (6.4) |
| Total lymphocytes (x10^9) mean (median; SD) | 1334 (1100; 1394.34) | 1298.12 (1210; 705.96) | 1391.42 (1170; 1260.65) | 916.93 (1021.5; 1794.73) | 0.614 | 0.548 | 0.923 | 19 (6.1) |
| **Laboratory data at 30 days** | | | | | | | | |
| CRP (mg/L) mean (median; SD) | 28.52 (10; 40.37) | 44.857 (9; 55.59) | 19.48 (9; 31.89) | 31.08 (11.5; 40.87) | 0.177 | 0.406 | 0.543 | 35 (11.2) |
| Ferritin (mg/L) mean (median; SD) | 509.06 (366; 725.14) | 762.57 (694; 342.83) | 545.56 (213.5; 1074.0) | 378.02 (250.5; 363.45) | 0.610 | 0.537 | 0024* | 271 (86.9) |
| **Vaccination n (%)** | 35 (11.2) | 11 (21.2) | 14 (9.2) | 10 (9.3) | 0022* | 0.957 | 0039* | 0 (0) |
| **Inpatient treatment n (%)** | | | | | | | | |
| Corticosteriods | 300 (96.5) | 42 (80.8) | 152 (99.3) | 106 (100) | 0.000** | 0.404 | 0.000** | 1 (0.3) |
| Remdesivir | 43 (13.8) | 6 (11.5) | 25 (16.3) | 12 (11.2) | 0.404 | 0.244 | 0.952 | 0 (0) |
| Tocilizumab | 23 (7.4) | 0 (0) | 7 (4.6) | 16 (15) | 0.117 | 0.004** | 0.003** | 0 (0) |
| Heparin | 311 (99.7) | 52 (100) | 153 (100) | 106 (99.1) | Not applicable | 0.857 | 1.000 | 0 (0) |
| **Length of hospital stay Median (IQR) days** | 13 (9–19) | 8 (7–12.5) | 12 (9–16) | 17 14(24) | 0,002** | 0.000** | 0.000** | 2 (0.6) |
| **Total dose Anakinra (mg) Mean** | 985.90 | 925.00 | 977.78 | 1027.10 | 0.563 | 0.481 | 0.295 | 0 (0) |
| **Days of Anakinra treatment Mean** | 13.42 | 11.02 | 13.54 | 14.39 | 0.239 | 0.645 | 0.135 | 2 (0.6) |

CRP: C-reactive protein. SaFi: SaO2/ FiO2. Relationship between transcutaneous oxygen saturation and fraction of inspired oxygen. CO-RADS: COVID-19 Reporting And Data System. CT: computed tomography. SCOPE: Severe COvid Prediction Estimate score.

**Table 4. Clinical evolution at 30 days in the different treatment groups.**

| CLINICAL EVOLUTION AT 30 DAYS | Group 1 | Group 2 | Group 3 | p-value (Groups 1–2) | p-value (Groups 2–3) | p-value (Groups 1–3) |
|---|---|---|---|---|---|---|
| | n (%) | n (%) | n (%) | | | |
| Discharge | 38 (73.1%) | 103 (67.3%) | 60 (56.6%) | <0.05 | <0.05 | <0.05 |
| ICU | 1 (1.9%) | 3 (2%) | 1 (0.9%) | N/A | N/A | N/A |
| Death | 11 (21.2%) | 41 (26.8%) | 29 (27.4%) | < 0.05 | N/A | <0.05 |
| Inpatient | 2 (3.8%) | 6 (3.9%) | 17 (16%) | N/A | <0.05 | <0.05 |

ICU: Intensive Care Unit.

## Clinical outcomes

The percentage of cure at 30 days was higher in patients who started anakinra in the first two days of admission compared to those who started after the fifth day (73.1% vs. 56.6%). In the latter, a higher percentage of patients were still hospitalized on day 60 (16%) with a significantly longer mean hospital stay (p-value = 0.001) (Table 4, Figs 3–6).

At 60 days, 221 (70.8%) were discharged, 42 (13.5%) required ICU admission, 87 (27.8%) had died and 4 (1.4%) remained hospitalized.

The mean dose of anakinra administered was 1000 mg (100–2600 mg) with significant differences in the mean total dose and clinical outcome. The dose of anakinra was higher in those patients who were discharged without being admitted to the ICU compared to those who required admission to the ICU or died (p-value = 0.000).

In addition, there were statistically significant differences between the mean value of a "simplified" SCOPE score and the clinical outcome at 60 days. The mean value of this score was lower in patients discharged without requiring ICU admission compared to those who died or required ICU admission (6.32 vs. 6.83 and 6.91; P-value = 0.020) (Fig 7).

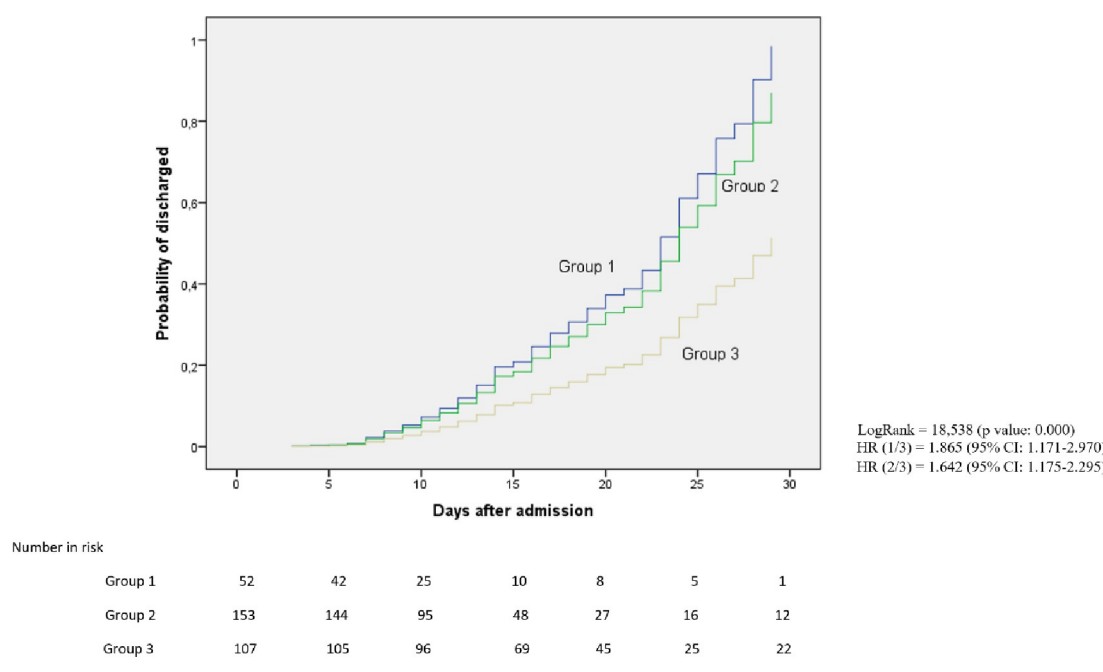

**Fig 3. Probability of medical discharge at 30 days in the different treatment groups.**

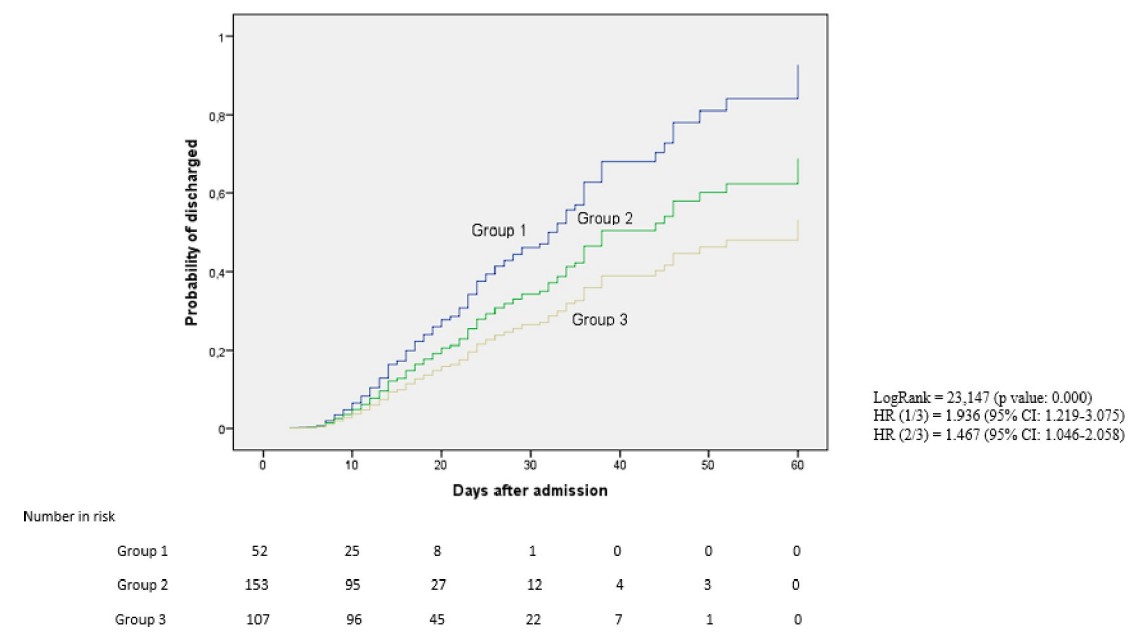

**Fig 4. Probability of medical discharge at 60 days in the different treatment groups.**

## Discussion

The sample size of 312 patients in the current single-center cohort study is the largest reported to date for patients with moderate/severe SARS-CoV-2 pneumonia treated with anakinra. The results showed better effectiveness when anakrina was administered early in the first two days

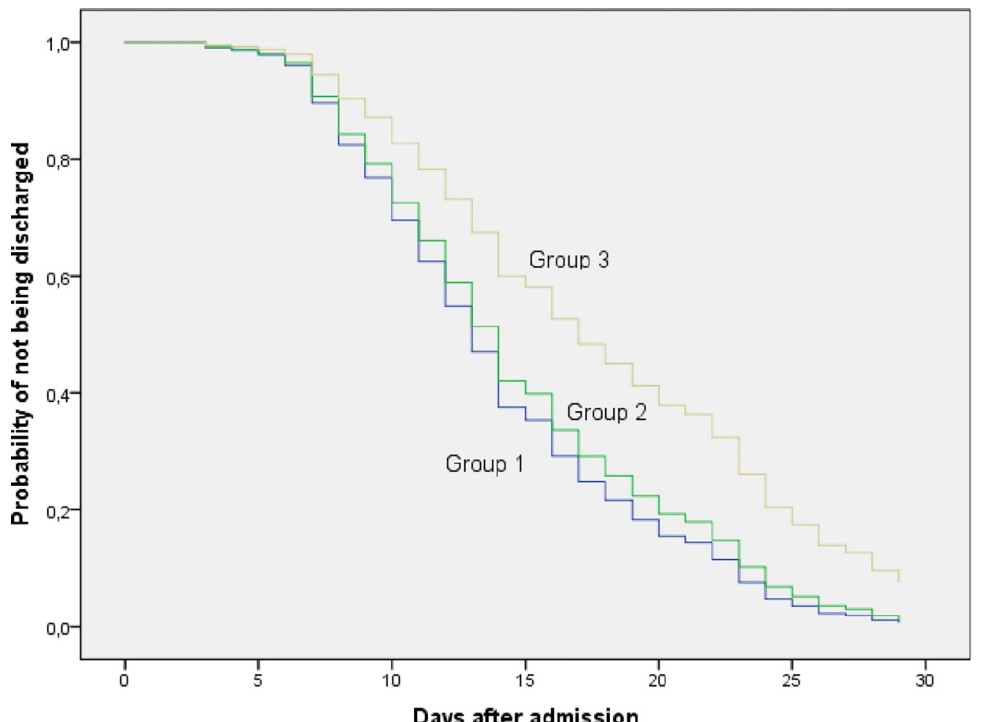

**Fig 5. Probability of remaining as an inpatient at 30 days in the different treatment groups.**

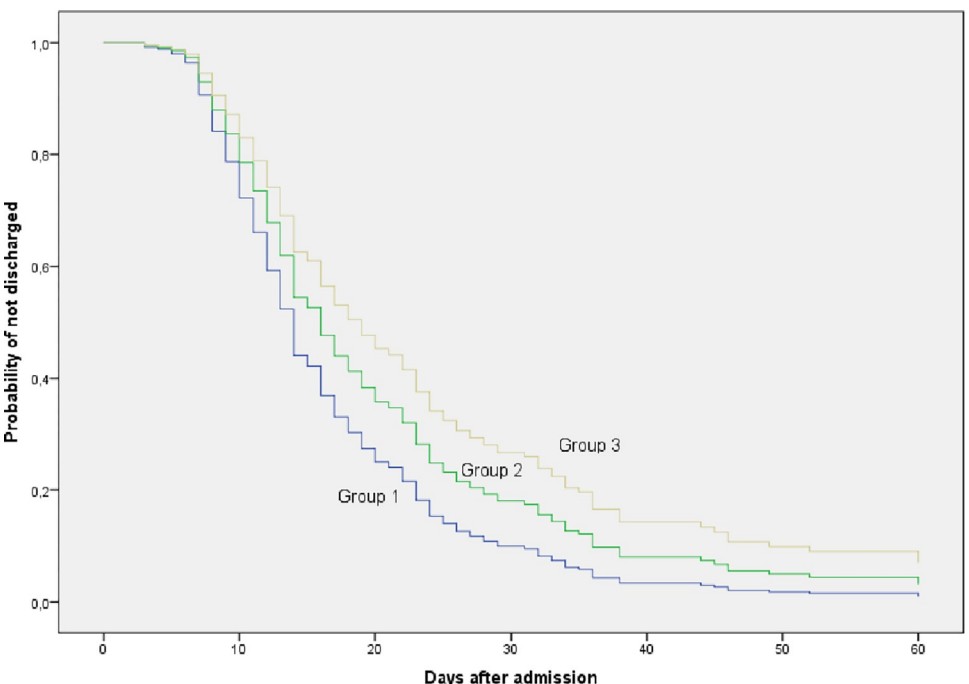

**Fig 6. Probability of remaining as an inpatient at 60 days in the different treatment groups.**

of admission, mainly in those with risk of progression to respiratory failure or death guided by a "simplified" SCOPE score. Although vaccination against SARS-CoV-2 is an important tool to reduce the spread of the virus and protect against severe disease [24, 25], there are

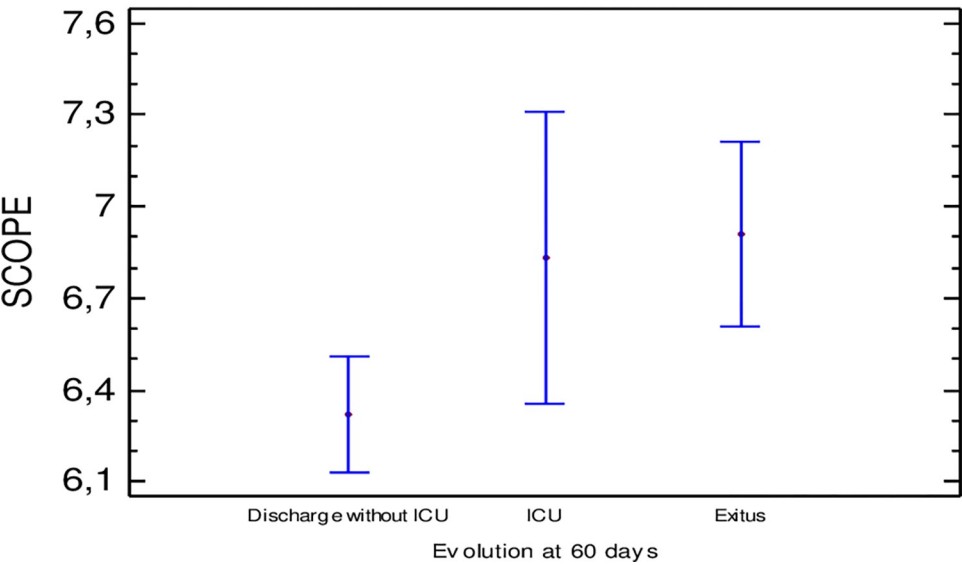

**Fig 7. Relationship between a simplified SCOPE score at start of treatment with anakinra and clinical outcome at 60 days.**

vaccinated patients who, for various reasons, still develop pneumonia with hyperinflammatory syndrome. In our cohort, the benefit of treatment with anakinra was obtained in both vaccinated and unvaccinated patients, although the percentage of those who were vaccinated was small—only 35 patients (11.2%) in our cohort- given that the end date of data collection was on 31 January 2022.

Regarding age and comorbidities, the data from our cohort are in agreement to those described in different observational studies carried out in Europe, where patients admitted for moderate/severe pneumonia due to SARS-CoV-2 are mainly men around 70 years of age and with cardiovascular disease as the main comorbidity [26, 27].

Severe cases of COVID-19 are characterized by a dysregulated inflammatory response that can ultimately lead to respiratory failure and death. Circulating alarmins, of critically ill COVID-19 patients induces tissue-specific inflammatory responses through an IL-1-mediated mechanism. This could be attenuated through inhibition of IL-1 receptor or of IL-1α [28].

The involvement of inflammasomes in the dysregulated inflammation in patients with SARS-CoV-2 infection has been demonstrated due to the activation of key inflammatory molecules such as IL-1. The inflammasome responsible for the synthesis of IL-1 consists of the NLRP3 receptor protein, ASC and caspase-1 and is activated in response to SARS-CoV-2 infection. Caspase-1 is activated by proteolytic scission and promotes the activation of substrates, including the inflammatory cytokines IL-1β and IL-18 involved in the onset of the inflammation cascade in these patients. These inflammasome derivatives are correlated with COVID-19 severity markers such as IL-6, CRP and LDH [29].

The urokinase-type plasminogen activator (uPA) receptor (uPAR) is a receptor primarily expressed in cells involved in immune responses -peripheral blood mononuclear cells, neutrophils and endothelial cells-. Its soluble receptor (suPAR) is involved in several immune functions, including cell adherence, migration, chemotaxis, proteolysis, immune activation, tissue remodeling, invasion and the transduction of inflammatory signals playing a key role in the physiopathology of ARDS associated with COVID-19 infection (inflammation, endothelial activation and coagulation). Its serum concentrations are stable throughout the day and are not influenced by fasting. Elevated concentrations of suPAR have been linked to poor clinical outcome in patients with various infections [30]. suPAR concentrations rise earlier in disease progression than other biomarkers -CRP, IL-6, ferritin and DD- and are indicative of dangerous associated molecular patterns (DAMPs) such as calprotectin (S100A8/A9) and IL-1α which contribute to pathogenic inflammation in COVID-19. It has been shown that calprotectin stimulates the aberrant production of IL-1β by circulating monocytes and that the inhibition of Il-1α prevents the development of strong pro-inflammatory responses [31].

Prior to the COVID-19 pandemic, several studies had already demonstrated an important role of suPAR in the early risk assessment of patients with sepsis, with serum levels being higher in patients with the worst clinical outcome [32]. In patients infected with SARS COV-2, high plasma concentrations of suPAR provide an early warning of the activation of inflammatory and coagulation pathways as well as of the endothelial-neutrophil interaction before the development of clinical signs and symptoms of inflammation, therefore their determination may stratify early those patients at risk of progression to severe respiratory failure or death. However, the determination of suPAR concentrations is not carried out in many hospitals and an equivalent predictive measure called Score of Estimation of Serious COVID Prediction (SCOPE) derived from circulating concentrations of CRP, D-dimers, IL-6 and ferritin has recently been developed and validated. A SCOPE score of more than 6 points has been shown to be comparable to a suPAR concentration of ≥6 ng/ml in terms of prognosis, thus it is an alternative to suPAR to predict progression to respiratory failure or death in patients with COVID 19 pneumonia and can be used to guide treatment decisions [17].

For all of the above, it can be stated that the circulating concentration of suPAR predicts progression to respiratory failure or death by reflecting several underlying biological processes that play an important role in inflammation. In the event that its determination is not possible, a SCOPE score of more than 6 points would be an alternative to predict progress to respiratory failure or death in patients with SARS-CoV-2 infection.

Anakinra is a recombinant IL-1 receptor antagonist that is commonly used to treat hyperinflammatory conditions and, therefore its use in the hyperinflammatory phase of moderate/severe SARS-CoV2 infection improves clinical symptoms and survival by blocking IL-1α and IL-1β controlling the altered regulation of cytokine responses [10, 28].

The SAVE-MORE clinical trial [16] conducted in 594 patients with COVID-19 at risk of progressing to respiratory failure based on plasma suPAR ≥6 ng/mL showed a better clinical outcome in patients treated with anakinra compared to placebo. The probability of a worse clinical outcome at day 28 with anakinra compared with placebo was 0.36 (95% CI: 0.26–0.50), and the proportion of patients who fully recovered exceeded 50%. In our cohort of patients treated with anakinra and corticosteroids, 64.4% had recovered by one month and 70.8% by two months. The laboratory follow-up of the SAVE MORE patients showed a significant increase in the absolute lymphocyte count on day 7 of treatment with a decrease in CRP levels, which is also consistent with our observations. Although the SAVE-MORE clinical trial used fixed doses of anakinra (100 mg/d/10 days), other studies have used different dosages, some adjusted for weight, and there is no consensus regarding the optimal dose of anakinra [33, 34]. In our study, we observed that those patients who had received a higher cumulative dose of anakinra had a better clinical outcome.

The dose used in our center was higher than that of the SAVE MORE clinical trial, as we administered 100 mg every 12 hours until clinical improvement, followed by 100 mg/day for a minimum of 5 days and continuing based on clinical course. We observed that some patients required these higher doses to achieve clinical improvement, likely due to a higher level of inflammation and not having been stratified early by the suPAR or SCOPE.

As it is not possible to determine the suPAR in most hospitals, a validated score called SCOPE has been developed whose correlation with suPAR is around 75% and which is calculated based on the plasma levels of CRP, ferritin, DD and IL-6 [17]. Although several studies have pointed to a correlation between elevated serum IL-6 levels and the development of respiratory failure in patients with COVID-19 [35–37], the lack of routine availability of this measure in many hospitals led us to use the SCOPE without taking IL6 levels into account -called a simplified SCOPE- and to assess its behavior in our study. In our cohort, the patients with the lowest values of a "simplified" SCOPE (Fig 5) similar to suPAR and SCOPE had a better clinical outcome.

## Conclusions

To our knowledge, this is the largest clinical practice cohort study of patients with moderate/severe SARS-CoV-2 pneumonia treated with anakinra that describes the clinical characteristics and course, taking into account the patient's vaccination status. Of the 312 patients analyzed, 202 (64.75%) were discharged by one month and 221 (70.8%) by two months, which supports the effectiveness of this strategy.

The best clinical outcome was observed in those patients who received anakinra early in the first two days of admission and had a better SaFi, likely due to early blockage of the cytokine storm—"window of opportunity"-. Of note, the clinical response to early treatment with anakinra was independent of the concomitant treatment administered, and possibly vaccination status.

In our cohort, the value of a simplified SCOPE score was higher in patients with a worse clinical course. Thus, we postulate that in the absence of suPAR and IL6 levels, a simplified SCOPE could be used as an estimator of the risk of progression to severe respiratory failure and serve as a guide for the early use of anakinra.

Our data support early treatment with anakinra together with corticosteroids in patients with moderate/severe SARS-CoV-2 pneumonia and hyperinflammation regardless of vaccination and concomitant treatment.

## Supporting information

**S1 File. Statistical analysis of all assessed variables in our cohort and among different treatment groups.**
(PDF)

## Acknowledgments

We thank all the health personnel who have worked hard throughout the pandemic and Ingrid de Ruiter for her translation of the manuscript into English.

Likewise, we also thank Dr. Jesús Candel Fábregas for his tireless struggle to improve our health service.

## Author Contributions

**Conceptualization:** Ismael Francisco Aomar-Millán, Úrsula Torres-Parejo, Laura Pérez Fernández, Silvia Martínez-Diz, Angel Ceballos Torres, Juan Salvatierra.

**Data curation:** Ismael Francisco Aomar-Millán, Javier Martínez de Victoria-Carazo, Daniel Fernández Reyes, Úrsula Torres-Parejo, Laura Pérez Fernández, Silvia Martínez-Diz, Angel Ceballos Torres, Jairo López Gómez, Francesco Bizzarri, Enrique Raya Álvarez, Juan Salvatierra.

**Formal analysis:** Úrsula Torres-Parejo.

**Funding acquisition:** Ismael Francisco Aomar-Millán, Juan Salvatierra.

**Investigation:** Ismael Francisco Aomar-Millán, Úrsula Torres-Parejo, Laura Pérez Fernández, Silvia Martínez-Diz, Angel Ceballos Torres, Juan Salvatierra.

**Methodology:** Ismael Francisco Aomar-Millán, Úrsula Torres-Parejo, Silvia Martínez-Diz, Juan Salvatierra.

**Resources:** Ismael Francisco Aomar-Millán, Juan Salvatierra.

**Software:** Úrsula Torres-Parejo.

**Supervision:** Ismael Francisco Aomar-Millán, Úrsula Torres-Parejo, Juan Salvatierra.

**Validation:** Ismael Francisco Aomar-Millán, Úrsula Torres-Parejo, Juan Salvatierra.

**Visualization:** Ismael Francisco Aomar-Millán, Javier Martínez de Victoria-Carazo, Daniel Fernández Reyes, Úrsula Torres-Parejo, Laura Pérez Fernández, Silvia Martínez-Diz, Angel Ceballos Torres, Jairo López Gómez, Francesco Bizzarri, Enrique Raya Álvarez, Juan Salvatierra.

**Writing – original draft:** Ismael Francisco Aomar-Millán, Úrsula Torres-Parejo, Juan Salvatierra.

**Writing – review & editing:** Ismael Francisco Aomar-Millán, Juan Salvatierra.

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
