## [Decision Letter · Decision Letter 0]

30 Jan 2023

PONE-D-22-32933CHARACTERISTICS AND CLINICAL OUTCOME IN 312 PATIENTS WITH MODERATE TO SEVERE PNEUMONIA DUE TO SARS-COV-2 AND HYPERINFLAMMATION TREATED WITH ANAKINRA AND CORTICOSTEROIDS: A RETROSPECTIVE COHORT STUDY.

PLOS ONE

Dear Dr. Aomar-Millan,

Thank you for submitting your manuscript to PLOS ONE. After careful consideration, we feel that it has merit but does not fully meet PLOS ONE’s publication criteria as it currently stands. Therefore, we invite you to submit a revised version of the manuscript that addresses the points raised during the review process.

We look forward to receiving your revised manuscript.

Kind regards,

Elisa Martín-Montañez

Academic Editor

PLOS ONE

Journal Requirements:

   "The authors have received funding from SOBI for the translation into English of the present manuscript." 

   "- Ismael F. Aomar Millán has participated on Advisory boards of Ferrer Internacional, Sanofi, SOBI, Astra Zeneca, and GSK.

- Juan Salvatierra has participated on Advisory boards and Presentations of Roche, SOBI, Novartis, Lilly, GSK, and Pfizer.

- Enrique Raya has participated on Advisory boards and received study funds from Novartis, Roche, Lilly, GSK, and Pfizer.

The remaining authors declare no conflict of interests."

Additional Editor Comments:

This study with a sample size of 312 patients with moderate/severe SARS-CoV-2 pneumonia treated with anakinra is the largest single-center cohort study reported to date. Due to the type of study the authors should comfort the observational studies guideline. Please consult, check the STROBE Statement - checklist of items that should be included in reports of observational studies and modify the manuscript.

The results show that in clinical practice patients achieved better outcomes when anakinra is administered early in the first two days of admission, and mainly in those with risk of progression to respiratory failure guided by a “simplified” SCOPE score. The anakinra early administration and the use of this simplified score are the main results of this manuscript and both should be clearly included in the research question.

Please, rewrite the objective in “abstract”, where is not included the assess of clinical outcome based on the time after hospital admission, but however, the conclusion is about this aim. And change the following text: “the simplified SCOPE score” by “a simplified SCOPE score”.

Please rewrite the research question/objective in “Introduction” considering as the main objective to assess whether there were differences in the clinical outcome based on the time after hospital admission of anakinra administration and the possible usefulness of a "simplified" SCOPE score to stratify the risk of worse prognosis in patients.

Regarding the SCOPE score, in “Introduction” the authors should clarify the use of IL-6 instead of IL-1.

And regarding the “simplified” SCOPE score proposed, the authors should explain how they create this SCOPE score in “Material and methods”.

“Review 2” Comments are written in Spanish. The reviewer consider the manuscript should include the mechanism of action of the molecule anakinra and why is used in this patients, in order to improve the manuscript understanding.

Reviewers' comments:

Reviewer's Responses to Questions

**Comments to the Author**

1. Is the manuscript technically sound, and do the data support the conclusions?

Reviewer #1: Yes

Reviewer #2: Yes

2. Has the statistical analysis been performed appropriately and rigorously? 

Reviewer #1: Yes

Reviewer #2: Yes

3. Have the authors made all data underlying the findings in their manuscript fully available?

Reviewer #1: Yes

Reviewer #2: Yes

4. Is the manuscript presented in an intelligible fashion and written in standard English?

Reviewer #1: Yes

Reviewer #2: Yes

5. Review Comments to the Author

Reviewer #1: This is a retrospective study to assess the effect and the timing of anakinra administration on clinical outcomes in a cohort of patients with moderate/severe SARS-CoV-2 pneumonia, and the simplified SCOPE score as a predictor for assessing response effective treatment. The authors found that earlier treatment with anakinra and corticosteroids was associated with better outcomes, showing also that the simplified SCOPE was a good prognostic tool. The article is interesting and well written. I have, however, some comments for the authors.

In particular:

- In the discussion section authors should add a brief description of the pathophysiological mechanisms supporting the use of anakinra in the Covid-19 setting (see also Renieris, G. et al. IL-1 mediates tissue specific inflammation and severe respiratory failure in Covid-19: clinical and experimental evidence.).

- Authors declared in the discussion section “Although several studies have pointed to a strong correlation between elevated serum IL-6 levels and the development of respiratory failure in patients with COVID-19 (31), the role of IL-6 during viral lung infections is still debated. Furthermore, several experimental models of viral lung infections have shown that IL-6 can have both pathogenic (32) and protective (33) effects. For these reasons, and the lack of routine availability in many hospitals, we decided to use this prognostic tool regardless of serum IL-6 levels to assess its behavior in our study.”.

Anakinra, however, is an IL-1 inhibitor. I suggest that the authors revise the entire section and the associated references to focus on the IL-1 pathway and suPAR (rather than IL-6) and describe the SCOPE score as an alternative method respect to the determination of suPAR levels for treatment decisions.

Minor revision

- Authors should check the representation of figure 3 and figure 4 (LogRank, HR, HR should not to be underlined).

- References are listed in a different style. Authors should review and correct them.

Reviewer #2: En mi opinión el artículo adolece, supongo que en aras de la brevedad, de no explicar al menos someramente qué es, cúal es el mecanismo y el porqué del uso de anakinra en estos paciente. Para el lector no versado sería importante. Son cuatro líneas que harán, sin duda, más comprensible el buen trabajo presentado.

6. PLOS authors have the option to publish the peer review history of their article (what does this mean?). If published, this will include your full peer review and any attached files.

Reviewer #1: **Yes: **Federico Biscetti

Reviewer #2: **Yes: **FRANCISCO JAVIER PEREZ FRIAS

---

## [Author Response · Author response to Decision Letter 0]

1 Mar 2023

Corrections re.no. PONE-D-22-32933

Response to Reviewers:

We have ensured that the manuscript meets PLOS ONE´s style requirements.

We have included in the manuscript that the data were anonymized, so the ethical committee did not require informed consent. 

 "The authors have received funding from SOBI for the translation into English of the present manuscript." 

This information has been included in the manuscript.

 "- Ismael F. Aomar Millán has participated on Advisory boards of Ferrer Internacional, Sanofi, SOBI, Astra Zeneca, and GSK.

- Juan Salvatierra has participated on Advisory boards and Presentations of Roche, SOBI, Novartis, Lilly, GSK, and Pfizer.

- Enrique Raya has participated on Advisory boards and received study funds from Novartis, Roche, Lilly, GSK, and Pfizer.

The remaining authors declare no conflict of interests."

Please confirm that this does not alter your adherence to all PLOS ONE policies on sharing data and materials, by including the following statement: "This does not alter our adherence to PLOS ONE policies on sharing data and materials.” (as detailed online in our guide for authors http://journals.plos.org/plosone/s/competing-interests).

This information has been included in the manuscript.

If there are restrictions on sharing of data and/or materials, please state these. Please note that we cannot proceed with consideration of your article until this information has been declared. 

The minimal data set have been included as supplementary material.

In the cover letter we state the changes made to our data availability statement and that the minimal data set have been included as supplementary material.

We have reviewed our reference list and ensured that it is correct.

Additional Editor Comments:

This study with a sample size of 312 patients with moderate/severe SARS-CoV-2 pneumonia treated with anakinra is the largest single-center cohort study reported to date. Due to the type of study the authors should comfort the observational studies guideline. Please consult, check the STROBE Statement - checklist of items that should be included in reports of observational studies and modify the manuscript.

We have filled in the STROBE list, and it is attached as a supplementary file.

The results show that in clinical practice patients achieved better outcomes when anakinra is administered early in the first two days of admission, and mainly in those with risk of progression to respiratory failure guided by a “simplified” SCOPE score. The anakinra early administration and the use of this simplified score are the main results of this manuscript and both should be clearly included in the research question.

Please, rewrite the objective in “abstract”, where is not included the assess of clinical outcome based on the time after hospital admission, but however, the conclusion is about this aim. And change the following text: “the simplified SCOPE score” by “a simplified SCOPE score”.

Please rewrite the research question/objective in “Introduction” considering as the main objective to assess whether there were differences in the clinical outcome based on the time after hospital admission of anakinra administration and the possible usefulness of a "simplified" SCOPE score to stratify the risk of worse prognosis in patients.

The objective has been rewritten and included in the manuscript following your recommendations.

Regarding the SCOPE score, in “Introduction” the authors should clarify the use of IL-6 instead of IL-1.

The Severe Covid Prediction Estimate (SCOPE) is calculated from circulating concentrations of CRP, DD, ferritin and IL-6 to offers a predictive accuracy for progression to severe respiratory failure. IL1 is not included and therefore it can´t be used for its calculation. In the introduction the following paragraph has been added: 

“Recently, the Severe Covid Prediction Estimate (SCOPE) calculated from circulating concentrations of CRP, DD, ferritin and IL-6 has been validated to stratify the risk of progression to severe respiratory failure or death with a similar predictive accuracy to suPAR [17]. The SCOPE establishes a cut-off point of 7 points to stratify patients at risk of progressing to severe respiratory failure or death”.

And regarding the “simplified” SCOPE score proposed, the authors should explain how they create this SCOPE score in “Material and methods”.

Since the determination of IL-6 is not routinely available in most hospitals, we decided to apply the SCOPE without taking this variable into account and call it as “simplified” SCOPE. A “simplified” SCOPE score was calculated based on CRP, ferritin, and D-dimer levels.

The SCOPE score establishes a cut-off point of 7 points to stratify those patients at risk of progressing to severe respiratory failure.

This is explained in our manuscript on pages 6-7 and figure 1.

“Review 2” Comments are written in Spanish. 

Comments in Spanish have been translated into English. 

The reviewer considers the manuscript should include the mechanism of action of the molecule anakinra and why is used in this patients, in order to improve the manuscript understanding.

The following paragraph has been included in the introduction of the manuscript:

“Anakinra is a recombinant IL-1 receptor antagonist that is commonly used to treat hyperinflammatory conditions and, therefore its use in the hyperinflammatory phase of moderate/severe SARS-CoV2 infection improves clinical symptoms and survival by blocking IL-1α and IL-1β controlling the altered regulation of cytokine responses [10]”. 

Reviewers' comments:

Reviewer's Responses to Questions

Comments to the Author

1. Is the manuscript technically sound, and do the data support the conclusions?

Reviewer #1: Yes

Reviewer #2: Yes

2. Has the statistical analysis been performed appropriately and rigorously?

Reviewer #1: Yes

Reviewer #2: Yes

3. Have the authors made all data underlying the findings in their manuscript fully available?

The PLOS Data policy requires authors to make all data underlying the findings described in their manuscript fully available without restriction, with rare exception (please refer to the Data Availability Statement in the manuscript PDF file). The data should be provided as part of the manuscript or its supporting information or deposited to a public repository. For example, in addition to summary statistics, the data points behind means, medians and variance measures should be available. If there are restrictions on publicly sharing data—e.g. participant privacy or use of data from a third party—those must be specified.

Reviewer #1: Yes

Reviewer #2: Yes

4. Is the manuscript presented in an intelligible fashion and written in standard English?

Reviewer #1: Yes

Reviewer #2: Yes

5. Review Comments to the Author

Reviewer #1: This is a retrospective study to assess the effect and the timing of anakinra administration on clinical outcomes in a cohort of patients with moderate/severe SARS-CoV-2 pneumonia, and the simplified SCOPE score as a predictor for assessing response effective treatment. The authors found that earlier treatment with anakinra and corticosteroids was associated with better outcomes, showing also that the simplified SCOPE was a good prognostic tool. The article is interesting and well written. I have, however, some comments for the authors.

In particular:

- In the discussion section authors should add a brief description of the pathophysiological mechanisms supporting the use of anakinra in the Covid-19 setting (see also Renieris, G. et al. IL-1 mediates tissue specific inflammation and severe respiratory failure in Covid-19: clinical and experimental evidence.).

The following paragraphs has been included in the manuscript: 

“Anakinra is a recombinant IL-1 receptor antagonist that is commonly used to treat hyperinflammatory conditions and, therefore its use in the hyperinflammatory phase of moderate/severe SARS-CoV2 infection improves clinical symptoms and survival by blocking IL-1α and IL-1β controlling the altered regulation of cytokine responses [10].”

“Severe cases of COVID-19 are characterized by a dysregulated inflammatory response that can ultimately lead to respiratory failure and death. Circulating alarmins, of critically ill COVID-19 patients induces tissue-specific inflammatory responses through an IL-1-mediated mechanism. This could be attenuated through inhibition of IL-1 receptor or of IL-1α [28]”.

- Authors declared in the discussion section “Although several studies have pointed to a strong correlation between elevated serum IL-6 levels and the development of respiratory failure in patients with COVID-19 (31), the role of IL-6 during viral lung infections is still debated. Furthermore, several experimental models of viral lung infections have shown that IL-6 can have both pathogenic (32) and protective (33) effects. For these reasons, and the lack of routine availability in many hospitals, we decided to use this prognostic tool regardless of serum IL-6 levels to assess its behavior in our study.”.

Anakinra, however, is an IL-1 inhibitor. I suggest that the authors revise the entire section and the associated references to focus on the IL-1 pathway and suPAR (rather than IL-6) and describe the SCOPE score as an alternative method respect to the determination of suPAR levels for treatment decisions.

In the discussion the paragraph referring the role of IL6 in viral lung infections have been deleted and we have focused on the IL-1 pathway and suPAR and describe the SCOPE score as an alternative method to the determination of suPAR levels for treatment decisions.

The following paragraphs have been added in the discussion:

Severe cases of COVID-19 are characterized by a dysregulated inflammatory response that can ultimately lead to respiratory failure and death. Circulating alarmins, of critically ill COVID-19 patients induces tissue-specific inflammatory responses through an IL-1-mediated mechanism. This could be attenuated through inhibition of IL-1 receptor or of IL-1α [28]. The involvement of inflammasomes in the dysregulated inflammation in patients with SARS-CoV-2 infection has been demonstrated due to the activation of key inflammatory molecules such as IL-1. The inflammasome responsible for the synthesis of IL-1 consists of the NLRP3 receptor protein, ASC and caspase-1 and is activated in response to SARS-CoV-2 infection. Caspase-1 is activated by proteolytic scission and promotes the activation of substrates, including the inflammatory cytokines IL-1β and IL-18 involved in the onset of the inflammation cascade in these patients. These inflammasome derivatives are correlated with COVID-19 severity markers such as IL-6, CRP and LDH [29]. The urokinase-type plasminogen activator (uPA) receptor (uPAR) is a receptor primarily expressed in cells involved in immune responses -peripheral blood mononuclear cells, neutrophils and endothelial cells-. Its soluble receptor (suPAR) is involved in several immune functions, including cell adherence, migration, chemotaxis, proteolysis, immune activation, tissue remodeling, invasion and the transduction of inflammatory signals playing a key role in the physiopathology of ARDS associated with COVID-19 infection (inflammation, endothelial activation and coagulation). Its serum concentrations are stable throughout the day and are not influenced by fasting. Elevated concentrations of SuPAR have been linked to poor clinical outcome in patients with various infections [30]. SuPAR concentrations rise earlier in disease progression than other biomarkers -CRP, IL-6, ferritin and DD- and are indicative of dangerous associated molecular patterns (DAMPs) such as calprotectin (S100A8/A9) and IL-1α which contribute to pathogenic inflammation in COVID-19. It has been shown that calprotectin stimulates the aberrant production of IL-1β by circulating monocytes and that the inhibition of Il-1α prevents the development of strong pro-inflammatory responses [31]. Prior to the COVID-19 pandemic, several studies had already demonstrated an important role of suPAR in the early risk assessment of patients with sepsis, with serum levels being higher in patients with the worst clinical outcome [32]. In patients infected with SARS COV-2, high plasma concentrations of suPAR provide an early warning of the activation of inflammatory and coagulation pathways as well as of the endothelial-neutrophil interaction before the development of clinical signs and symptoms of inflammation, therefore their determination may stratify early those patients at risk of progression to severe respiratory failure or death. However, the determination of suPAR concentrations is not carried out in many hospitals and an equivalent predictive measure called Score of Estimation of Serious COVID Prediction (SCOPE) derived from circulating concentrations of CRP, D-dimers, IL-6 and ferritin has recently been developed and validated. A SCOPE score of more than 6 points has been shown to be comparable to a suPAR concentration of ≥6 ng/ml in terms of prognosis, thus it is an alternative to suPAR to predict progression to respiratory failure or death in patients with COVID 19 pneumonia and can be used to guide treatment decisions [17]. For all of the above, it can be stated that the circulating concentration of suPAR predicts progression to respiratory failure or death by reflecting several underlying biological processes that play an important role in inflammation. In the event that its determination is not possible, a SCOPE score of more than 6 points would be an alternative to predict progress to respiratory failure or death in patients with SARS-CoV-2 infection.

Anakinra is a recombinant IL-1 receptor antagonist that is commonly used to treat hyperinflammatory conditions and, therefore its use in the hyperinflammatory phase of moderate/severe SARS-CoV2 infection improves clinical symptoms and survival by blocking IL-1α and IL-1β controlling the altered regulation of cytokine responses [10,28]. 

Minor revision

- Authors should check the representation of figure 3 and figure 4 (LogRank, HR, HR should not be underlined).

Figures 3 and 4 have been corrected by removing the underline.

- References are listed in a different style. Authors should review and correct them.

We have reviewed all references, and all are listed in the same style.

Reviewer #2: En mi opinión el artículo adolece, supongo que, en aras de la brevedad, de no explicar al menos someramente qué es, cúal es el mecanismo y el porqué del uso de anakinra en estos paciente. Para el lector no versado sería importante. Son cuatro líneas que harán, sin duda, más comprensible el buen trabajo presentado.

In the introduction and discussion, paragraphs have been added to make the rationale for the use of anakinra in these patients more understandable, as well as its mechanism of action.

6. PLOS authors have the option to publish the peer review history of their article (what does this mean?). If published, this will include your full peer review and any attached files.

Do you want your identity to be public for this peer review? For information about this choice, including consent withdrawal, please see our Privacy Policy.

Reviewer #1: Yes: Federico Biscetti

Reviewer #2: Yes: FRANCISCO JAVIER PEREZ FRIAS

---

## [Decision Letter · Decision Letter 1]

12 Mar 2023

CHARACTERISTICS AND CLINICAL OUTCOME IN 312 PATIENTS WITH MODERATE TO SEVERE PNEUMONIA DUE TO SARS-COV-2 AND HYPERINFLAMMATION TREATED WITH ANAKINRA AND CORTICOSTEROIDS: A RETROSPECTIVE COHORT STUDY.

PONE-D-22-32933R1

Dear Dr. Aomar-Millan,

We’re pleased to inform you that your manuscript has been judged scientifically suitable for publication and will be formally accepted for publication once it meets all outstanding technical requirements.

Kind regards,

Elisa Martín-Montañez

Academic Editor

PLOS ONE

Additional Editor Comments (optional):

Reviewers' comments:

Reviewer's Responses to Questions

**Comments to the Author**

1. If the authors have adequately addressed your comments raised in a previous round of review and you feel that this manuscript is now acceptable for publication, you may indicate that here to bypass the “Comments to the Author” section, enter your conflict of interest statement in the “Confidential to Editor” section, and submit your "Accept" recommendation.

Reviewer #1: All comments have been addressed

Reviewer #2: (No Response)

2. Is the manuscript technically sound, and do the data support the conclusions?

Reviewer #1: (No Response)

Reviewer #2: Yes

3. Has the statistical analysis been performed appropriately and rigorously? 

Reviewer #1: (No Response)

Reviewer #2: Yes

4. Have the authors made all data underlying the findings in their manuscript fully available?

Reviewer #1: (No Response)

Reviewer #2: Yes

5. Is the manuscript presented in an intelligible fashion and written in standard English?

Reviewer #1: (No Response)

Reviewer #2: Yes

6. Review Comments to the Author

Reviewer #1: (No Response)

Reviewer #2: (No Response)

7. PLOS authors have the option to publish the peer review history of their article (what does this mean?). If published, this will include your full peer review and any attached files.

Reviewer #1: **Yes: **Federico Biscetti

Reviewer #2: **Yes: **Javier Pérez Frías

---

## [Editor Report · Acceptance letter]

16 Mar 2023

PONE-D-22-32933R1 

Characteristics and clinical outcome in 312 patients with moderate to severe pneumonia due to SARS-COV-2 and hyperinflammation treated with anakinra and corticosteroids: A retrospective cohort study 

Dear Dr. Aomar-Millán:

I'm pleased to inform you that your manuscript has been deemed suitable for publication in PLOS ONE. Congratulations! Your manuscript is now with our production department. 

Kind regards, 

on behalf of

Dr. Elisa Martín-Montañez 

Academic Editor

PLOS ONE